# Production of a Dulaglutide Analogue by Apoptosis-Resistant Chinese Hamster Ovary Cells in a 3-Week Fed-Batch Process

**DOI:** 10.3390/ph18121896

**Published:** 2025-12-16

**Authors:** Rolan R. Shaifutdinov, Maria V. Sinegubova, Ivan I. Vorobiev, Polina E. Prokhorova, Alexey B. Podkorytov, Nadezhda A. Orlova

**Affiliations:** 1Laboratory of Mammalian Cell Bioengineering, Skryabin Institute of Bioengineering, Federal State Institution, “Federal Research Centre ‘Fundamentals of Biotechnology’ of the Russian Academy of Sciences”, 60 let Oktjabrja pr-t 7, bld. 1, 117312 Moscow, Russia; urtmyakowski@yandex.ru (R.R.S.); mvsineg@gmail.com (M.V.S.); ptichman@gmail.com (I.I.V.); 2Medsintez Plant LLC, Kirova 28, 620028 Ekaterinburg, Russia; polina_krop@mail.ru (P.E.P.); alexey-podkoritov@yandex.ru (A.B.P.)

**Keywords:** dulaglutide, GLP-1-Fc, CHO cells, transgene amplification, biosimilar, fed-batch, in vitro bioactivity

## Abstract

**Background:** Dulaglutide, a GLP-1-IgG4 Fc fusion, is a long-acting GLP-1 receptor agonist used for type 2 diabetes therapy and other emerging indications. It is produced commercially in Chinese hamster ovary (CHO) cells. The supply of the original drug is now limited in some regions, so creation of highly productive biosimilar manufacturing platforms is important. **Methods:** Two expression plasmids (p1.1-Tr2-Dul, p1.2-GS-Dul) encoding dulaglutide were sequentially transfected into apoptosis-resistant CHO 4BGD cells. Two-step transgene amplifications with methotrexate (MTX), followed by methionine sulfoximine (MSX) selection and subsequent cell cloning pipeline, were employed. Candidate clonal cell lines were selected using fed-batch culturing and long-term productivity testing. **Results:** Transfection with a second plasmid encoding glutamine synthetase (p1.2-GS-Dul) and selection with MSX resulted in a further ~30% increase titer in polyclonal population even after MTX-driven amplification. Top clone 4BGD/Dul #73 reached 1.05 g/L product titer in fed-batch culture (qP up to 22 pg·cell^−1^·day^−1^) and remained stable for 69 days in medium without MTX/MSX. Size exclusion-high-performance liquid chromatography showed ≥95% monomer; EC_50_ of the purified GLP-1-Fc in a GLP-1R/CRE-Luc assay was 52 pM for the obtained product versus 76 pM for the original reference drug. **Conclusions:** The sequential transfection and dual-marker selection approach enables the efficient generation of a robust, high-yield, and glutamine-independent CHO producer, representing a productive strategy suitable for industrial biosimilar development.

## 1. Introduction

Glucagon-like peptide-1 (GLP-1) is a 31-residue incretin released from enteroendocrine L-cells immediately after nutrient ingestion, binding to a class-B G-protein-coupled receptor expressed on pancreatic β-cells, the central nervous system, and several peripheral targets. Interaction with GLP-1R stimulates glucose-dependent insulin secretion, suppresses glucagon release, delays gastric emptying, and reduces appetite, thereby lowering glycemia without provoking hypoglycemia and simultaneously promoting weight loss. Consequently, human GLP-1 receptor agonists have become first-line pharmacotherapeutics for type 2 diabetes and obesity whenever at least one weight-related comorbidity is present [1,2]. Native GLP-1 is inactivated by dipeptidyl-peptidase-4 (DPP-4); its plasma half-life does not exceed two minutes [3]. To circumvent this limitation, long-acting analogues were developed—biosynthetic fusion proteins such as dulaglutide [4] and albiglutide [5]. In dulaglutide, the GLP-1(7–37) analogue containing the stabilizing substitutions Gly8, Glu22, and Gly36 is genetically fused through a flexible (Ser_4_Gly)_3_ linker to the Fc domain of human IgG4 (Figure 1A), and albiglutide utilizes human serum albumin as the half-life-extending carrier. Both variants allow once-weekly administration, which is more convenient than twice-daily exenatide or once-daily liraglutide administration. Nevertheless, albiglutide was withdrawn in 2017 due to suboptimal clinical efficacy in diabetes treatment.

Dulaglutide (CAS 923950-08-7) was approved by the US FDA in 2014 and registered in Russia in 2016. The secreted molecule is a covalent dimer; the mature dimer is stabilized by two intra-chain and two inter-chain disulfide bridges, and carries a single N-glycan at Asn126. Commercial manufacturing is based on Chinese hamster ovary (CHO) cells in chemically defined suspension culture [6].

Industrial CHO platforms are frequently derived from the DXB11 or DG44 sublines, both lacking functional DHFR (dihydrofolate reductase) and therefore suitable for methotrexate (MTX)-driven gene amplification. Published dulaglutide bioanalogue production systems utilize diverse CHO backgrounds—from early CHO K1 and CHO S derivatives requiring serum supplementation [7] to engineered CHO K1 SV-KO lines achieving 1.5 g/L product titer in premium FortiCHO medium under CMV promoter control [8]; however, such viral promoters often suffer from silencing during extended culture. Contemporary bioprocesses demand parental hosts exhibiting fast growth, genetic stability, and resistance to detrimental stresses in high-density fed-batch culture. To address these requirements, we previously created the CHO 4BGD line by multiplex CRISPR/Cas9 editing of bak1, bax, dhfr, and glul genes, followed by stable overexpression of Bcl-2 and Beclin-1 with the endogenous EEF1A1 promoter [9,10]. Simultaneous inactivation of apoptosis mediators BAK1/BAX and metabolic markers DHFR and GS (glutamine synthetase) conferred apoptosis resistance, enabled dual-marker selection, prolonged culture longevity, and supported elevated integral viable cell densities in fed-batch mode.

We have previously developed the p1.1 plasmids based on the Chinese hamster EEF1A1 promoter–intron–terminator unit linked to concatemerized Epstein–Barr virus terminal repeats (EBVTRs), which stimulate chromosomal integration and facilitate high-copy amplification under MTX selection [11]. The bicistronic design couples the target open reading frame with a murine DHFR marker via an attenuated EMCV IRES. This vector enabled high-yield production of blood-clotting factors VIII [12] and IX [13], and follicle-stimulating hormone [14]. The truncated p1.1-Tr2 variant lacks non-essential sequences, preserves productivity while increasing the genome integration rate [15], and was successfully used for SARS-CoV-2 S-protein RBD and human ACE2-Fc production for clinical diagnostic purposes [16,17].

We assumed that combining the apoptosis- and metabolism-optimized CHO 4BGD host with the pair of highly stable EEF1A1 promoter-based vector plasmids, bearing DHFR and GS selection markers, could be used for rapid-producer cell development.

## 2. Results

### 2.1. Design of Expression Constructs

A dual-vector strategy was selected to maximize transcriptional output and subsequent genomic amplification of the dulaglutide gene. The first construct, p1.1-Tr2-Dul, was assembled by inserting a 905 bp synthetic open reading frame encoding GLP-1-Fc into the p1.1-Tr2 backbone. The expression cassette incorporates (i) the signal peptide of the mouse Ig κ light chain (METDTLLLWVLLLWVPGSTG, Ig kappa chain V-III region MOPC 63) [18], (ii) a consensus Kozak context (GCCGCCACCATGG) immediately upstream of the start codon, and (iii) an attenuated EMCV IRES that links the dulaglutide ORF to a downstream DHFR marker enabling MTX selection (Figure 1B). To provide an orthogonal selection marker, the same dulaglutide ORF was sub-cloned into p1.2-GS vector, yielding p1.2-GS-Dul plasmid (Figure 1B). This vector plasmid contains the glutamine-synthetase (GS) selection marker gene, driven by the separate SV40 promoter, and is not intended for genomic amplification.

**Figure 1 pharmaceuticals-18-01896-f001:**
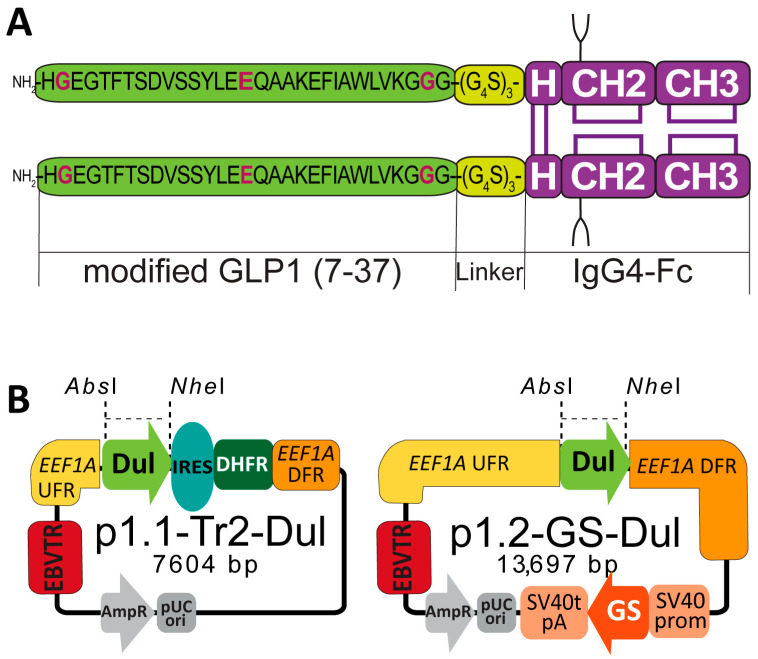
Scheme of the GLP-1-Fc and expression plasmids used. (**A**) Scheme of the GLP-1-Fc molecule. Each monomer consists of the modified GLP-1(7–37) peptide (mutations highlighted in magenta), fused via a (G4S)_3_ flexible linker to the human IgG4 Fc domain. The Fc region includes the hinge (H) and constant domains CH2 and CH3. Two identical chains form a covalent homodimer through intra- and inter-chain disulfide bonds. (**B**) Functional maps of the expression plasmids. The dulaglutide ORF (Dul) is inserted between AbsI and NheI restriction sites (shown in italics). The expression is driven by the eukaryotic translation elongation factor-1 alpha (EEF1A1) promoter–intron–terminator expression cassette. In p1.1-Tr2-Dul (7604 bp), the EEF1A1 upstream and downstream flanking regions (UFR/DFR) are truncated. The p1.1-Tr2-Dul construct contains an attenuated EMCV IRES followed by the murine DHFR selection marker gene; the p1.2-GS-Dul plasmid contains the SV40 promoter, rat glutamine synthase ORF (GS), and SV40 terminator and polyadenylation signal (SV40 polyA). Both plasmids include EBVTR—fragment of the long terminal repeat from the Epstein–Barr virus, replication origin (pUC ori) and ampicillin resistance gene with the corresponding promoter (AmpR).

### 2.2. Generation and Amplification of a DHFR-Selectable Pool

CHO 4BGD cells were nucleofected with p1.1-Tr2-Dul and subjected to primary selection in the presence of 200 nM MTX. Within 14 days, viability recovered to >90%, and an ELISA-measured target protein titer of 3.5 ± 0.1 µg mL^−1^ was obtained for a 3-day culture, which corresponded to the specific productivity 1.3 pg cell^−1^ day^−1^ (Figure 2A). Two further MTX-driven amplification steps (2 µM and 8 µM) were conducted, each followed by a 12–20-day recovery. Stepwise MTX selection increased volumetric titer approximately nine-fold to 30.1 ± 0.4 µg mL^−1^ (specific productivity of 12.9 pg cell^−1^ day^−1^, Figure 2A).

To surpass the MTX-driven productivity plateau, the 8 µM MTX-amplified cell pool was re-transfected with p1.2-GS-Dul plasmid and selected using 25 µM MSX and supporting 8 µM MTX for 24 days. A further 60% increase in specific productivity up to 18.5 pg cell^−1^ day^−1^ was detected for the double-selected cell pool (Figure 2A), while cell division rate was not reduced upon selection and amplification (Figure 2B). Thus, sequential insertion of the GS-linked vector after MTX amplification provided a substantial productivity boost while rendering the cells glutamine-autonomous.

### 2.3. Pilot Fed-Batch Performance of the Dual-Selected Pool

The obtained cell pool, designated as 4BGD/p1.1-Tr2-Dul(+)GS, was tested for sustainability of product secretion in fed-batch conditions. In Eden B100S medium supplemented with F100aS/F100bS feeds, polyclonal cell pool reached 11 × 10^6^ cells mL^−1^ (Figure 3A) and accumulated 499 ± 11 µg mL^−1^ dulaglutide (Figure 3B) in a 12-day run. The use of ActiFroxx medium with Feed 1/Feed 2 allowed us to achieve lower cell densities (Figure 3A) and lower titers of 411 ± 43 µg mL^−1^ (Figure 3B) under similar conditions. Cell population remained viable until the culture termination, but the product titer increase was visible for only 8 to 10 days, probably due to depletion of highly productive cell subpopulation in the dense culture.

### 2.4. Clonal Cell Lines Generation

The polyclonal cell pool 4BGD/pTr2-Dul(+)GS was used to obtain clonal lines, and a total of 160 single colonies were obtained from 1728 seeded wells of 96-well plates, resulting in the cloning efficiency of 10%. All 160 colonies were transferred into 24-well plates, grown for 4 days, and screened for dulaglutide titers. Forty-two best-producing clones were scaled up to 6-well plates and screened again. Twelve clones with the maximum titer were transferred to 125 mL Erlenmeyer flasks and tested for product titer, growth rate, and cell morphology in the 3-day suspension batch culture. Clones #73 and #105 were top ranked due to having the highest productivity (qP 14–17 pg·cell^−1^·day^−1^, Figure 4A) and the lowest doubling time (25 h, Figure 4B); therefore, they were selected for further experiments in fed-batch process, along with the other four best-dividing clones.

### 2.5. Fed-Batch Culturing of Clonal Lines

Comparative analysis of clonal lines #73, #104, #105, #114, #137, and #141 revealed significant medium-dependent performance differences between the clones. Adaptation of six selected clones to the EmCD CHO 101 medium, suitable for fed-batch culturing, was unsuccessful for two clones (#114 and #141). One clone, #137, was unable to grow in fed-batch conditions for more than 6 days, and one other clone, #104, dropped the specific productivity by approximately twofold. Two remaining clones—#73 and #105—demonstrated superior productivity in this medium, achieving peak titers of 1047 mg/L and 952 mg/L, respectively (Figure 5), when cultured with temperature shift to 34 °C and lower titer when cultured at 37 °C (Figure 6). Notably, both of these clones, but not the low-producing clone, #104, maintained > 90% viability until day 15 in EmCD CHO 101 medium, and lowered the lactate level to 20 mM at the day 6, eliminating lactate-associated stress. Accumulation of ammonia ions was low for all three clones; thus, it may not be the cause of productivity drop for the #104 clone.

Adaptation of the same clones to the other fed batch-suitable medium, SagiCHO, was successful in all cases, but three clones dropped the specific productivity by at least threefold—the same clones that were unable to grow in EmCD CHO 101 medium. Clones #73, #105, and #105 demonstrated enhanced cell proliferation rates, achieving approximately 30% higher peak cell densities compared to EmCD CHO 101 cultures (18.7 vs. 14.3 × 10^6^ cells/mL for #73 clonal line). However, this improved growth phenotype did not translate to increased productivity, with SagiCHO cultures yielding 2-fold lower final titers (520 vs. 1047 mg/L for clone #73) (Figure 6). Ammonium accumulation exceeded the 10 mM threshold earlier in SagiCHO (day 9–11 vs. day 11–15 in EmCD CHO 101), coinciding with growth plateau and titer stagnation (Figure 5 and Figure 6).

Temperature shift was essential for high titer development in the EmCD CHO 101, and clones #73 and #105 gave 30% titer drop when cultured at 37 °C; the peak cell density remained the same for both culture conditions, and the main cause of lower-end titer during the 37 °C run was the earlier drop of cell density and viability (Figure 7). Clone #73′s scalability was confirmed in 120 mL shake-flask cultures, reinforcing EmCD CHO 101 as the optimal medium for this clone. Amino acid analysis of the spent medium revealed that, for both clones #73 and #105, all residual amino acid concentrations were maintained at or above optimal levels, eliminating amino acid nutrients depletion as a limiting factor for dulaglutide production (Appendix A).

The selected best-performing clone, #73, was also tested in a few other fed batch-suitable media, utilizing the Erlenmeyer flasks (Figure 8) and TPP microbioreactors (Figure 9). Earlier cell viability decline was also seen for most media tested in the constant 37 °C cultivation, but the end titer was higher for the CDM34 medium at 37 °C (Figure 9), so the temperature downshift is beneficial for only some culture media variants and detrimental for others. In all cases, the end-titer was sometimes lower than in the EmCD CHO 101 medium. Microbioreactor cultures gave lower peak cell density, shorter culture duration, and sometimes lower end-titer, probably due to higher shear stress.

### 2.6. Productivity Stability

Two clonal cell lines, #73 and #105, were tested for the long-term specific productivity dynamics during the periodic cultivation without toxic selection agents (MTX and MSX). A significant decrease in specific productivity and increase in the cell densities in 3-day cultures were detected for both clones (Figure 10A,C), making the basal EmCD CHO 101 medium unsuitable for cell bank generation and seed culture preparation during the commercial dulaglutide production. This unexpected productivity drop may be caused by the presence in the medium of hypoxanthine and thymidine (HT); these compounds allowed the free division of the DHFR-negative cells, which silenced or eliminated most copies of the target gene with the DHFR marker. Long-term cultivation was repeated in ProCHO5 medium, which contains no HT. No significant decrease in specific productivity or final product titer was observed for both clones during 69 days of periodic batch culturing (Figure 10B,D).

Genomic DNA from the clone #73 was subjected to the NGS analysis. No mutations were detected in the dulaglutide ORF. Overall plasmid copy number was calculated as 219+7 copies of the dulaglutide-containing plasmids (DHFR+GS) per one diploid cell.

### 2.7. Target Protein Quality and Bioactivity

At the end of fed-batch culturing in EmCD CHO 101 medium of clone #73, dulaglutide was purified usingthe Protein A affinity chromatography and diafiltered to the 10 mM sodium citrate pH 6.5 solution for further analysis. SDS-PAGE revealed that the purified dulaglutide exhibited indistinguishable electrophoretic mobility, 70 kDa for dimers and 37 kDa for monomeric forms, compared to the original dulaglutide drug substance (Figure 11). No proteolytic degradation products were detected in both reducing and non-reducing conditions, demonstrating absence of target-product proteolytic degradation during the 18-day fed-batch culture. Both the test sample and reference have the same position as the main band on isoelectric focusing. Size exclusion HPLC (SEC-HPLC) showed 96.8% of monomeric dulaglutide (in SEC terminology, this corresponds to the native covalent dimer of GLP-1-Fc) in the test sample and 97.5% monomeric form in the Trulicity sample; both samples have mainly the dimeric admixture forms (Figure 11).

RP-HPLC analysis indicated that test sample contains 0.96% acidic isoforms and 14.3% basic isoforms, and the Trulicity sample contains similar levels of these isoforms—0.8%/18.6% (Figure 11).

Initial analysis of the glycoforms was performed by intact mass spectrometry of the dulaglutide polypeptide chains. Nine glycoforms were identified in the test sample (Table 1), dominated by species at 31,286 Da (51%) and 31,448 Da (12%), corresponding to the G0F and G1F structures; these masses were also dominant in the reference product. In addition, the test sample contained two low-abundance species (31,366 Da and 31,528 Da) with a combined abundance of approximately 20%. The reference product exhibited one unique minor species (31,141 Da, ~3%). Both samples also contained low levels of the 31,489 Da glycoform, consistent with a bisected G0FB structure, and the 31,610 Da glycoform, corresponding to G2F. Such differences in minor glycoforms reflect the inherent microheterogeneity of CHO N-glycosylation and are commonly observed across Fc-fusion proteins produced on different manufacturing platforms.

Peptide mapping with MS detection was performed for the test sample alone, reaching 95.4% coverage; all identified peptides matched the theoretical amino acid sequence of dulaglutide (Appendix A).

The dulaglutide derived from clone #73 was also tested for specific biological activity using HEK293 cells, stably transfected withthe GLP-1R and the luciferase reporter gene with the cAMP responsive element (CRE). The in vitro EC_50_ ratio (test/reference), measured by GLP-1R/CRE Luciferase Reporter HEK293 Cell Line, was 0.73, confirming sufficient specific bioactivity of the dulaglutide preparation in vitro (Figure 12).

## 3. Discussion

This study demonstrates that combining DHFR-based target gene amplification system and the GS-selection marker within a dual-auxotrophic apoptosis-resistant host cell line enables efficient cell line development without extensive clones screening and made possible achievement of gram-per-liter production titers within 18 days fed-batch culture. We reached a specific productivity (qP) of 22 pg cell^−1^ day^−1^—comparable to the highest values reported for autophagy-modulated cell line, expressing the hEPO-Fc target protein (qP 23 pg cell^−1^ day^−1^ at 32 °C) [19].

We used the sequential selection approach instead of co-transfection and double selection, thus eliminating a possible bottleneck in the start of the selection-amplification process. Transfection of target gene by the plasmid with GS marker to the genome-amplified cells allowed for 1.43-fold increase in qP, thus demonstrating that cells were not at the maximal possible specific productivity after the genomic amplification of the plasmid with DHFR selection marker. This four-step dual-marker strategy provides a balanced trade-off between development speed, genetic stability, and volumetric productivity, forming a robust and scalable foundation for future cost-efficient dulaglutide biosimilar development.

While various GLP-1-Fc-producing cell lines have been described with variable productivity rates, our approach addresses several critical limitations of existing Fc-fusion protein expression systems. Retroviral transduction approaches, though achieving comparable titers (3.15 g/L in 14-day fed-batch), raise significant safety concerns regarding wild-type retrovirus co-transfer [20].

A key advantage of our producer cell lines development method is high genetic stability; both selected clonal cell lines demonstrate constant specific productivity for >60 doublings without any selection pressure, bothwithout HT and glutamine supplementations, fully mimicking real seed train and bioreactor medium composition. Genomic analysis indicates that high-copy integration of target genes under the EEF1A1 promoter is compatible with long-term stability.

The clones cultivated in EmCD CHO 101 medium showed noticeable differences in growth dynamics and product accumulation, consistent with expected clone-specific variability in CHO production systems. As shown in Figure 5 and Figure 7, clones #73 and #105 were among the most productive variants within this dataset. The applied temperature shift to 34 °C also supported increased product accumulation, in line with commonly reported effects of moderate hypothermia on CHO cell productivity. As in many long fed-batch processes, the cultures eventually entered a decline phase characterized by a rapid drop in viability, most prominently in the 37 °C cultures. Although apoptosis-resistant cells typically maintain high viability for an extended period, they ultimately undergo a terminal collapse; the underlying mechanism is not yet fully understood and may involve a sudden loss of mitochondrial integrity; however, we have no direct evidence for this at present. For safety and consistency, the process was terminated when viability fell below 80%, but in some runs, the sudden viability decline began before the next scheduled measurement, resulting in visibly low-viability points on the graphs (Figure 7A, orange line). These observations illustrate both the robustness and the limits of the system developed.

In the context of bioprocess development, Critical Quality Attributes (CQAs) are the measurable product characteristics that must remain within acceptable limits to ensure safety and efficacy, whereas Critical Process Parameters (CPPs) are the operational conditions that may influence these CQAs. Because this work focuses on establishing a stable producer cell line rather than validating a manufacturing process, we did not undertake a formal Quality-by-Design (QbD) CPP–CQA mapping. Nevertheless, several development-stage CQAs relevant to dulaglutide quality were systematically monitored, including monomer content after Protein A purification (>95% by SEC-HPLC), absence of degradation fragments detected by SDS-PAGE, charge-variant distribution, glycoform composition preserving the dominant G0F/G1F species, and in vitro GLP-1R potency. Operational conditions such as temperature shift, feeding strategy, and lactate control were applied empirically to support culture longevity and productivity; however, at this stage, they were not designated as CPPs because no process optimization or validation studies were performed. This level of characterization is appropriate for early-stage analytical comparability and provides a foundation for more detailed CPP–CQA analysis in future development.

Functional characterization is an important part of the development of a biosimilar candidate. Cell-based in vitro assays are a viable alternative to in vivo assays for assessing whether a recombinant preparation demonstrates activity comparable to the reference product. Similar functional activity in such assays provides early evidence that the biosimilar may exhibit safety and efficacy profiles similar to the originator when used in animals or humans. Importantly, the bioactivity of the developed dulaglutide analogue was comparable to that of the originator product.

The two-marker selection approach described in this article contrasts favorably with conventional single-plasmid methods yielding 80–1000 mg/L in basic CHO hosts [21]; CMV-based expression platforms, which are prone to target protein expression loss; or technically challenging retroviral platforms that require special safety grade. Taken together, these findings establish a robust, scalable, and economically viable platform for dulaglutide biosimilar production that combines the advantages of high productivity and genetic stability—representing a significant advancement in therapeutic protein manufacturing that bridges the gap between academic innovation and industrial application.

## 4. Materials and Methods

### 4.1. Expression Vector Construction

The dulaglutide coding sequence was obtained as a synthetic DNA fragment (914 bp) containing adaptor sites for AbsI and NheI restriction enzymes (Sibenzyme, Novosibirsk, Russia) for cloning into the p1.1 family vectors. The synthetic sequence included the signal peptide from mouse immunoglobulin kappa chain light chain (METDTLLLWVLLLWVPGSTG) and the consensus Kozak sequence (GCCGATGG) for translation initiation. The fragment was initially cloned into pUC57 and subsequently transferred as a 905 bp AbsI-NheI fragment encoding dulaglutide into the p1.1-Tr2 expression vector [15]. Correct assembly and absence of mutations in p1.1-Tr2-Dul were confirmed by complete plasmid sequencing using Oxford Nanopore technology with analysis of read length distribution performed at Cloning Facility JSC, Moscow, Russia. The p1.2-GS-Dul construct was assembled by cloning the dulaglutide sequence via AbsI-NheI sites into the p1.2-GS vector [11]. Correct assembly and absence of mutations in the protein-coding sequence were verified by Sanger sequencing using primers SQ-5CH6-F and IRESArev (Appendix A Appendix A) performed at the Inter-Institutional Center for Collective Use “GENOME” IMB RAS using BigDye Terminator v. 3.1 cycle sequencing kit and ABI PRISM 3730 genetic analyzer (Applied Biosystems, Waltham, MA, USA). Plasmids for transfection were propagated in Escherichia coli TOP10 strain and purified using Plasmid Midiprep kit (Evrogen, Moscow, Russia). Plasmids were concentrated by ethanol precipitation: adding 1/10 volume of 3M sodium acetate and three volumes of 96% ethanol, centrifugation at 13,000 rpm for 10 min, washing with 70% ethanol, and aseptically dissolved in the sterile phosphate buffer R (Invitrogen, Carlsbad, CA, USA) overnight at 4 °C.

### 4.2. Cell Culture and Transfection

CHO 4BGD cells [10] were maintained in 125 mL Erlenmeyer flasks (Corning, One Riverfront Plaza, Corning, NY, USA) in ProCHO-5 medium (Lonza, Basel, Switzerland) supplemented with 8 mM L-glutamine,100 μM hypoxanthine, and 16 μM thymidine (all supplements from PanEco, Moscow, Russia) at 37 °C, under 5% CO_2_, with orbital shaking at 155 rpm (10 mm amplitude) in MCO-20AIC Sanyo incubator (Panasonic, Osaka, Japan). Cells were passaged 24 h before transfection by dilution with warm culture medium.

Ten million cells per transfection were pelleted, washed once with DPBS (PanEco, Moscow, Russia), and then resuspended in 100 μL SBB buffer (250 mM sucrose, 1 mM MgCl_2_, DPBS) containing 50 μg of precipitated plasmid DNA dissolved at 1–3 mg/mL concentration in the same buffer. Then, 2.5 micrograms of pEGFP-N2 reporter plasmid (Takara Bio Inc., Kusatsu, Shiga, Japan) was added to each transfection. Cells were electroporated using Invitrogen Neon Transfection System with Neon 100 μL transfection kit at 1700 V, single pulse, 20 ms, and immediately transferred to 30 mL warm ProCHO-5 medium supplemented with hypoxanthine–thymidine and glutamine. Cells were cultured for 48 h in non-selective medium before the selection.

### 4.3. Sequential Selection and Gene Amplification

Post-transfection selection was initiated by transferring cells to hypoxanthine–thymidine-free medium containing 200 nM MTX (Sandoz, Holzkirchen, Germany). Cells were passaged every 3–6 days until viability was restored to 90% (19 days). Sequential gene amplification was performed by stepwise increasing MTX concentration to 2 μM and 8 μM, with 12–15 days culture time at each concentration until 90% viability and constant doubling time were achieved. Cell viability was determined using Countess II automated cell counter (Invitrogen, USA) with 0.4% trypan blue staining.

For dual-marker selection, the MTX-amplified polyclonal population was re-transfected with p1.2-GS-Dul plasmid using identical electroporation conditions. Methionine sulfoximine (MSX, Sigma, Burlington, MA, USA) selection was applied at a concentration of 25 μM with maintenance of 8 μM MTX throughout the selection period in glutamine-free medium. After 22 days of culturing, the cell viability restored to 90% and initial growth rate.

### 4.4. Clone Isolation, Screening, and Upscaling

Clone isolation was performed via the limiting dilution method. Polyclonal cell pool stabilized at the selection pressure 8 uM MTX and 25 uM MSX was cultured for two passages in non-selective conditions. Cells were passaged 24 h before cloning. Cloning medium was prepared of 90% EmACF CHO 212 medium (Eminence, Suzhou, China) supplemented with 100 μM hypoxanthine, 16 μM thymidine, 4 mM glutamine, 4 mM alanyl-glutamine (PanEco, Moscow, Russia), and 10% conditioned medium from untransfected CHO culture. The target seeding density was 1 cell per well, 100 uL per well in 96 well-plates. Plates were incubated for fourteen days at 37 °C and 5% CO_2_ in a static incubator, and single growing colonies were identified using light microscopy.

For the most high-producing clones, upscaling was performed: upon reaching >70% confluence (after 4–7 days), colonies were transferred to 24-well plates diluted 1:5–1:10 with ProCHO5 medium, then to 6-well plates, and finally to shake flasks. At each stage, cell morphology and growth rate were evaluated, and productivity was measured using ELISA. Colonies demonstrating high expression levels were adapted to suspension culture in ProCHO-5 medium without selection agents.

### 4.5. Fed-Batch Cultivation

#### 4.5.1. Microbioreactor Experiments

Fed-batch cultivations in microbioreactors were performed in TPP 50 mL tubes with a working volume of 20 mL at 37 °C and orbital shaking at 240 rpm (orbital diameter 50 mm). Clonal cell lines to be evaluated were seeded at 0.5 or 1.0 × 10^6^ cells/mL depending on the experiment. The following media systems were evaluated: CHO CDM29, CHO CDM34 (all Tofflon, China, with supplementation by Tobitec FA and Tobitec FB feeds), Eden B100S with F100aS and F100bS feeds (Bioengine, Shanghai, China), and CHO Pro with SFF048P(A) and SFF048P(B) feeds (Himedia, Mumbai, Maharashtra, India). Beginning on day 3, feeds were added every 48 h, with Feed A at 4% (*v*/*v*) and Feed B at 0.4% (*v*/*v*) into each tube. Temperature was held constant at 37 °C throughout the entire cultivation period.

#### 4.5.2. Shake Flask Experiments

Fed-batch cultivation was additionally carried out in 30 mL Erlenmeyer flasks (Corning, USA) shaken at 170 rpm (orbital diameter 10 mm) at 37 °C. Cultivation media tested included EmCD CHO 101 with Feed A and Feed B (Eminence, Suzhou, China), SagiCHO (OPM Biosciences, Shanghai, China) with OPM-AF183 and Highly Concentrated Feed CDFS36, CHO CDM34 with Tobitec FA and Tobitec FB feeds (Tofflon, Shanghai, China), and ActiPro with Cell Boost 7a and Cell Boost 7b (Cytiva, Marlborough, MA, USA). Seeding density was 1.0 × 10^6^ cells/mL. For EmCD CHO 101, the standard feeding strategy entailed addition of 3% (*v*/*v*) Feed A and 0.3% (*v*/*v*) Feed B every 24 h, starting on day 3. For SagiCHO, the standard feeding strategy entailed addition of 5% (*v*/*v*) Feed A and 0.5% (*v*/*v*) Feed B every 48 h, starting on day 2. For ActiPro, the standard feeding strategy entailed addition of 2% (*v*/*v*) Feed A and 0.2% (*v*/*v*) Feed B every 24 h, starting on day 3. For CHO CDM34, the standard feeding strategy entailed addition of 3% or 4% (*v*/*v*) Feed A and 0.3 or 0.4% (*v*/*v*) Feed B every 48 h, starting on day 3. In some experiments on day 5, the temperature was reduced to 34 °C to extend culture longevity and increase specific productivity. Cultures were harvested after 18 days or when cell viability dropped below 80%.

#### 4.5.3. Process Monitoring and Sampling

Daily sampling of 300 μL was performed to evaluate viable cell density and metabolite content. Samples were centrifuged at 300× *g* for 5 min, and the supernatant stored at –20 °C for subsequent analysis. Glucose concentrations were monitored from day 3 with the Glucose biochemical kit (Diavettest, Russia), or with an OnCall Plus glucometer (Acon, San Diego, CA, USA). Culture medium was supplemented with sterile 20% glucose solution to maintain it in a range of 10–40 mM. Lactate quantification was performed on day 3, using the Lactic acid-Olvex diagnostic kit (Olvex Diagnosticum, Saint Petersburg, Russia). Ammonium levels were controlled from day 8 using the Ammonia kit (Diavettest, Pushchino, Russia). Dulaglutide concentrations in the supernatant were measured using an analytical Protein A column (BioCore, Nanochrome, Suzhou, China) starting on day 4. Amino acid depletion profiles were monitored using ion-exchange chromatography after post-column derivatization with ninhydrin. All cultures were maintained for 20 days or until viable cell density dropped below 50%.

### 4.6. Stability Studies

Stability testing was conducted by culturing the cells in the absence of selection pressure for up to 68–69 days. Cells were grown in EmCD CHO 101 medium, which contained hypoxanthine and thymidine, but not glutamine, in TPP tubes, with agitation at 240 rpm, or in ProCHO5 medium containing no hypoxanthine, thymidine, or glutamine in 30 mL Erlenmeyer flasks shaken at 160 rpm (10 mm orbital diameter). Every 3 or 4 days, cultures were split by diluting 1:6 to 1:10, resulting in the seeding density of 0.15 million cells/mL (for a four-day passage) or 0.3 million cells/mL (for a three-day passage). Viability, cell morphology, and productivity were tracked at every passage, and dulaglutide quantification in the culture medium was performed using Protein A HPLC. All analytical results were processed with statistical tools; significant differences were established using standard parametric tests, as implemented in the GraphPad QuickCalcs online calculator (GraphPad Software, LLC; URL: https://www.graphpad.com/quickcalcs/ttest1/; accessed on 15 September 2025).

### 4.7. Dulaglutide Purification

Conditioned medium was collected and separated from cells by centrifugation at 2000 rpm for 5 min, followed by clarification at 13,500 rpm for 10 min. Tris-HCl buffer (pH 8.0) was added to achieve a final concentration of 50 mM. The conditioned medium was loaded onto a 1 mL alkali-resistant Protein A column (NmTRAP, NanoMicro, Suzhou, China) pre-equilibrated with 50 mM Tris-HCl, pH 8.0.

The column was washed sequentially with 10 column volumes of high-salt buffer (1 M NaCl, 50 mM Tris-HCl, pH 8.0), followed by equilibration buffer (50 mM Tris-HCl, pH 8.0). Target protein was eluted using 50 mM sodium citrate buffer, pH 3.4. The eluted dulaglutide solution was immediately neutralized to pH 6.5 using Trizma base buffer (pH 9.0) and diafiltered toward 10 mM sodium citrate solution, pH 6.5, using centrifugal concentrators with PES membrane, cut-off 10 kDa (Biofil, Guangzhou, China).

### 4.8. Analytical Methods

**Size-exclusion chromatography (SEC-HPLC).** Product concentration and purity were assessed using a Superdex 200 10/300 GL column (Cytiva, Uppsala, Sweden) with phosphate-buffered saline mobile phase at 0.5 mL min^−1^ flow rate and UV detection at 280 nm. Sample injection volume was 20–50 μL. Column temperature was maintained at ambient conditions. Peak integration was performed using chromatography data system software to calculate percentages of monomer, dimer, and higher-molecular-weight species.

**Reverse-phase high-performance liquid chromatography (RP-HPLC).** Charge variant analysis was conducted using a ZORBAX 300SB-C3 column (250 mm, 300 Å pore size, Agilent Technologies, Santa Clara, CA, USA) with gradient elution from 40% to 60% B over 40 min at a 0.2 mL min^−1^ flow rate and UV detection at 215 nm. Mobile phase A consisted of 0.1% trifluoroacetic acid in water, and mobile phase B was 0.1% trifluoroacetic acid in acetonitrile. Sample preparation involved addition of acetonitrile to 5% final concentration and trifluoroacetic acid to 1% final concentration, followed by centrifugation at 13,500 rpm for 10 min.

**Protein A affinity chromatography for quantitative analysis.** Dulaglutide concentration was determined using analytical-scale Protein A chromatography. Samples were applied to BioCore Protein A column (Nanochrome, China), washed with equilibration buffer, and eluted with pH 3.6 sodium acetate buffer. Protein concentration was calculated based on UV absorbance at 280 nm, using theoretical extinction coefficient.

**Enzyme-linked immunosorbent assay (ELISA) for dulaglutide quantification.** Sandwich ELISA was performed using anti-human IgG antibodies (HEMA #XG36) for capture at 100 ng/well in PBS, incubated overnight at +4 °C. Wells were blocked with 3% BSA-PBS for 1 h at 37 °C, with shaking, at 300 rpm. Sample dilutions and calibration standards (Trulicity diluted in 1% BSA-PBS) were applied for 1 h at 37 °C. Detection was performed using HRP-conjugated anti-human IgG antibodies (HEMA #T271X) at 1:40,000 dilution. Color development used TMB substrate for exactly 10 min, stopped with 5% H_3_PO_4_, and measured at 450 nm using Feyond-A300 plate reader (Allsheng, Hangzhou, China).

**SDS–polyacrylamide gel electrophoresis (SDS-PAGE).** Analysis was performed using Mini-PROTEAN Tetra system (Bio-Rad, Hercules, CA, USA) with 12% acrylamide separating gels under reducing and non-reducing conditions. Samples were heated to 80 °C for 5 min with 6× sample buffer containing 10 mM dithiothreitol for reducing conditions. Gels were stained with colloidal Coomassie blue and analyzed using TotalLab TL120 densitometry software, v. 2009 (Nonlinear Dynamics, Newcastle Upon Tyne, UK).

**Isoelectric focusing (IEF).** Purified dulaglutide and the reference product were analyzed using the Mini IEF Cell system (Model 111) with EPS 600 power supply (Bio-Rad, USA). Polyacrylamide IEF gels were cast using a monomer–ampholyte mixture containing Bio-Lyte 3/10 ampholytes, polymerized under fluorescent light, and equilibrated prior to sample loading. Samples were desalted by threefold diafiltration through 10 kDa JetSpin centrifugal filters (Jet Biofil, Guangzhou, China), adjusted to 1% ampholytes and approximately 3 mg/mL protein. Aliquots of 0.5–2 µL were applied to the gel surface using a plastic template, and focusing was performed stepwise at 100 V for 15 min, 200 V for 15 min, and 450 V for 60 min. Gels were fixed in 12.5% trichloroacetic acid/30% methanol, stained with Coomassie R-250/CuSO_4_ staining solution, destained in ethanol–acetic-acid buffers, air-dried overnight, and scanned for analysis.

**Intact mass spectrometry.** Molecular weight determination was performed using Impact II Q-TOF mass spectrometer (Bruker Daltonik, Bremen, Germany) equipped with Apollo II electrospray ionization source and Elute UHPLC system. Samples were analyzed on YMC-Pack ODS-A column (5 μm, 2.1 × 250 mm, 300 Å) with 0.4 mL/min flow rate and 1:20 post-column flow splitting. Gradient elution: Isocratic 5% B for 5 min, and then 5% to 90% B over 15 min (A: 0.1% formic acid in water, B: 0.1% formic acid in acetonitrile). Column temperature was 40 °C, and injection volume was 15 μL. Ionization parameters: Capillary voltage, +4.5 kV; nebulizer gas, 0.7 bar; drying gas, 6 L/min at 180 °C; and scan range, m/z 500–3000 at 1 Hz. Internal calibration used sodium trifluoroacetate solution.

**Peptide mapping and glycosylation analysis.** Protein samples (≥10 μg) were denatured in 480 mg/mL urea and 0.7 mg/mL dithioerythritol in 50 mM ammonium bicarbonate pH 7.8, incubated for 1 h at 37 °C. Alkylation was performed with 29 mg/mL iodoacetamide for 30 min at room temperature in darkness. After 4-fold dilution with 50 mM ammonium bicarbonate, tryptic digestion was performed with sequencing-grade trypsin (Promega) at a 1: 50 protein/enzyme ratio for 16 h at 37 °C. LC-MS/MS analysis used Impact II Q-TOF system with Waters Acquity HSS T3 column (1.8 μm, 2.1 × 100 mm) at 35 °C. Gradient: 5% B for 3 min, 5% to 60% B over 37 min, 60% to 95% B over 5 min, hold at 95% B for 5 min. MS parameters: Positive ESI mode, scan range m/z 50–3000, full scan frequency 2 Hz, automatic MS/MS mode with 1.5–4 Hz dynamic frequency, and collision energy 55 eV at m/z 700 to 124 eV at m/z 1500.

**Bioactivity assay.** Biological activity was determined using GLP-1 receptor/cAMP response element–luciferase reporter cell line (GLP-1R/CRE Luciferase Reporter HEK293 Cell Line, BPS Bioscience, #78176, San Diego, CA, USA). Cells were seeded at 3 × 10^4^ cells per well in 96-well plates and incubated overnight. Serial dilutions of test samples and reference standard (Trulicity, batch D596262) were added in duplicate and incubated for 5 h at 37 °C, 5% CO_2_. Luciferase activity was measured following D-luciferin substrate addition (LuciFire™, MOLECTA, Moscow, Russia). EC_50_ values were calculated using four-parameter logistic regression implemented in the drc package (version 3.0-1) within the R statistical environment (version 4.5.1), and relative potency was determined by comparison to reference standard.

**Genomic analysis methods.** Whole-genome sequencing was performed using Illumina paired-end (150 bp). DNA libraries were prepared according to manufacturer protocols and sequenced to achieve >30× coverage. Reads were aligned to the Chinese hamster reference genome (CriGri-PICRH-1.0, GCA_003668045.2) using BWA algorithm. Transgene copy numbers were determined by the read’s coverage for the DHFR and GS ORF areas of plasmids, not resented in the host cell genome. Raw sequencing data are deposited at Sequence Read Archive, trace SRR36108296, BioProject PRJNA1366811.

All analytical assays were qualified according to internal research standard operation procedures, with verification of calibration linearity, repeatability, and instrument stability. This study did not aim to establish GMP-level validation; therefore, all analytical results should be interpreted within the context of research-grade method qualification.

## 5. Conclusions

This work establishes an efficient and streamlined framework for the rapid creation of genetically stable, high-performing CHO cell lines capable of producing a dulaglutide analogue at high titers in long fed-batch processes. The integrated genome-engineering strategy, which generated the apoptosis-resistant 4BGD background, combined with dual-marker selection, enabled robust culture performance, extended viability, and favorable metabolic behavior under moderate hypothermia. The resulting product showed high purity after Protein A capture, consistent glycan and charge-variant profiles dominated by G0F/G1F species, and in vitro GLP-1R potency comparable to the reference molecule.

The presented platform—spanning targeted genome construction, stable clone selection from a relatively small set of clones, and scalable fed-batch cultivation—provides a technically mature basis for further upstream and downstream intensification. The strategy is broadly applicable to other recombinant therapeutics and supports systematic exploration of feeding regimes, culture temperatures, and advanced analytical characterization.

Overall, this approach forms a strong technological foundation for the stepwise development of a dulaglutide biosimilar, combining genetic stability, process scalability, and favorable product quality attributes. Future work will extend this platform toward expanded analytical comparability and process refinement needed for later stages of biosimilar development.

## 6. Patents

The authors Rolan R. Shaifutdinov, Maria V. Sinegubova, Ivan I. Vorobiev, and Nadezhda A. Orlova are inventors on the Russian patent application (application number W25048729) covering the dulaglutide producer cell line.

## Figures and Tables

**Figure 2 pharmaceuticals-18-01896-f002:**
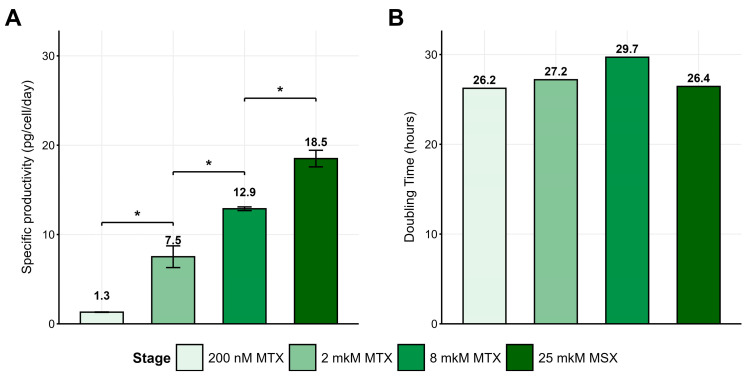
Dulaglutide specific productivity (**A**) and population-doubling time (**B**) for polyclonal line 4BGD/p1.1-Tr2-Dul during selection and amplification in ProCHO5 medium in the presence of increasing concentrations of MTX (200 nM, 2 μM, and 8 μM) and MSX (25 μM). Dulaglutide titer measured by ELISA, *n* = 2, two-tailed two-sample *t*-test, *—*p* < 0.05. Statistical significance should be interpreted with caution due to *n* = 2 technical replicates.

**Figure 3 pharmaceuticals-18-01896-f003:**
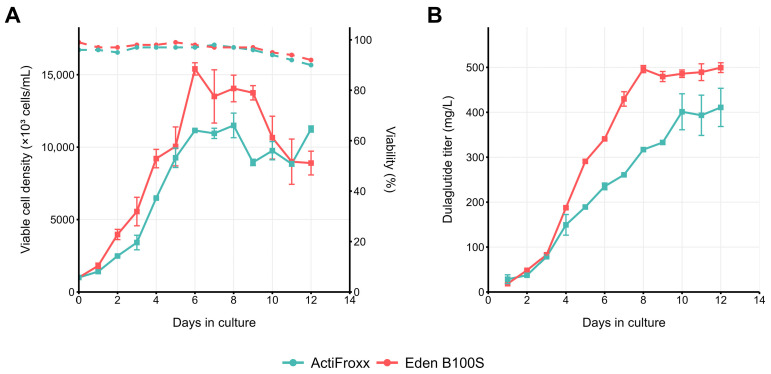
Fed-batch culturing of dulaglutide-expressing pool 4BGD/p1.1-Tr2-Dul(+)GS in 30 mL of different fed-batch media using 125 mL Erlenmeyer flasks under orbital shaking (20 mm diameter) at 37 °C. (**A**) Viable cell density (VCD, solid lines) and viability (dashed lines). (**B**) Dulaglutide titer accumulation. Dulaglutide titer measured by ELISA, n = 2.

**Figure 4 pharmaceuticals-18-01896-f004:**
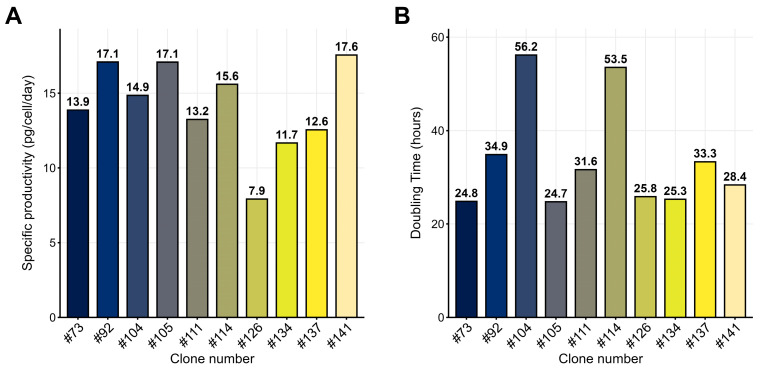
(**A**) Specific productivity and (**B**) doubling time for 10 clonal dulaglutide-producing lines adapted to suspension growth in a batch cycle, ProCHO5 medium. Dulaglutide titer measured by ELISA.

**Figure 5 pharmaceuticals-18-01896-f005:**
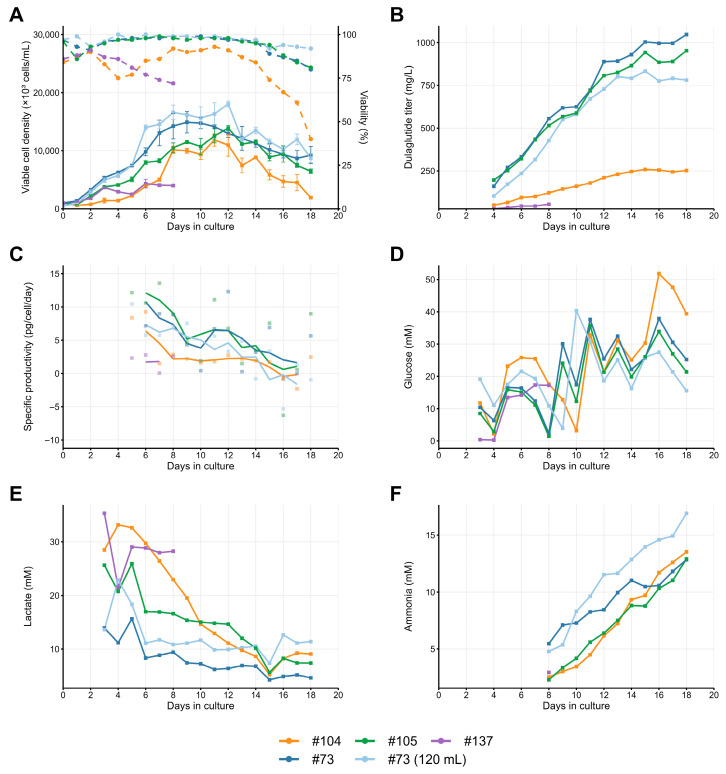
Fed-batch cultivation of dulaglutide-producing clonal cell lines in 30 mL of EmCD 101 basal medium using 125 mL Erlenmeyer flasks under orbital shaking (20 mm throw), with a temperature shift from 37 °C to 34 °C, on day 5. (**A**) VCD (solid lines) and viability (dashed lines). (**B**) Dulaglutide titer accumulation. (**C**) Specific productivity (qP) smoothed by 3 points. (**D**) Glucose concentration before feeding. (**E**) Lactate accumulation. (**F**) Ammonium level. Dulaglutide titer was measured by Protein A affinity chromatography.

**Figure 6 pharmaceuticals-18-01896-f006:**
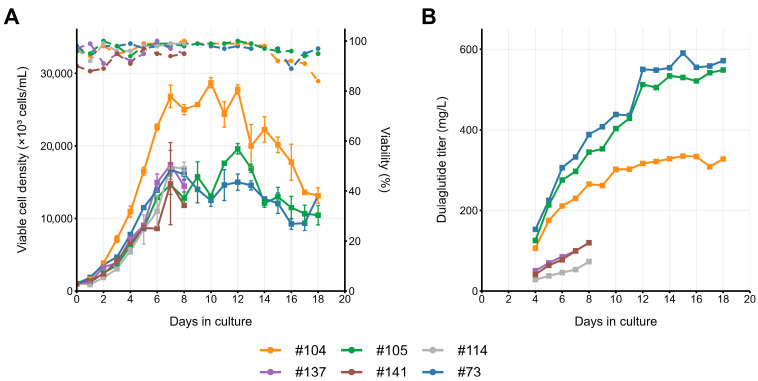
Fed-batch cultivation of six dulaglutide-expressing clonal cell lines, Erlenmeyer flasks, temperature shift to 34 °C on day 5, culture medium SagiCHO. (**A**) VCD (solid lines) and viability (dashed lines). (**B**) Dulaglutide titer. Dulaglutide titer measured by the Protein A analytical microcolumn.

**Figure 7 pharmaceuticals-18-01896-f007:**
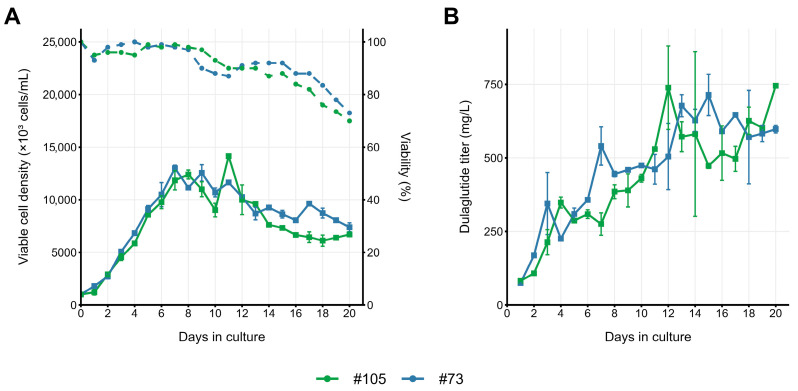
Fed-batch cultivation of two dulaglutide-expressing clonal cell lines in 30 mL of EmCD 101 basal medium using 125 mL Erlenmeyer flasks under orbital shaking (20 mm throw) at constant 37 °C. (**A**) VCD (solid lines) and viability (dashed lines). (**B**) Dulaglutide titer. Dulaglutide quantification was performed by ELISA, n = 2.

**Figure 8 pharmaceuticals-18-01896-f008:**
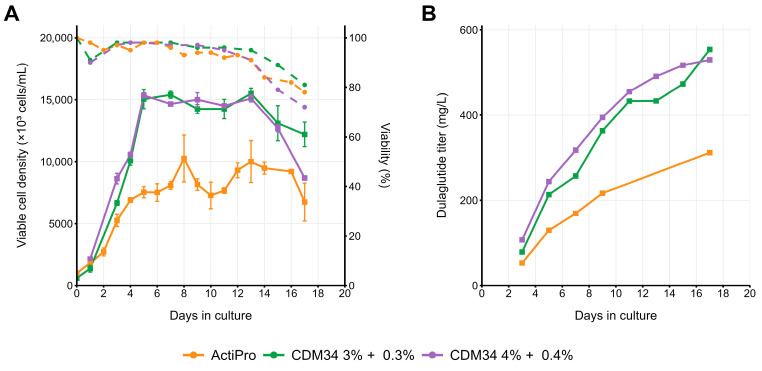
Fed-batch cultivation of clone #73 in various media, Erlenmeyer flasks, temperature shift to 34 °C on day 5. (**A**) VCD (solid lines) and viability (dashed lines). (**B**) Dulaglutide titer accumulation. Dulaglutide titer was measured by Protein A analytical microcolumn.

**Figure 9 pharmaceuticals-18-01896-f009:**
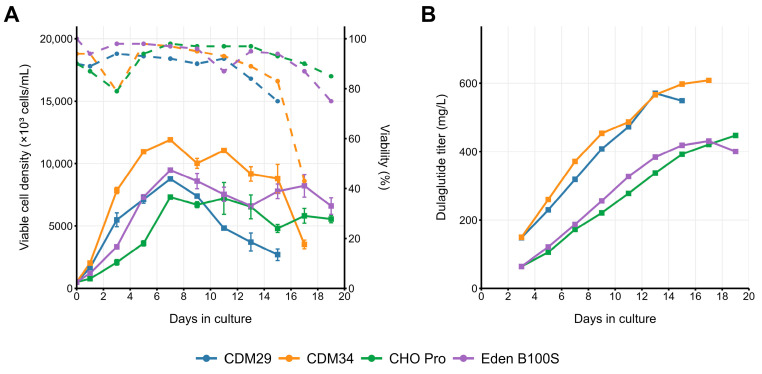
Fed-batch culturing of clone #73 in various media, TPP microbioreactors, constant temperature 37 °C. (**A**) VCD, solid lines and viability (dashed lines). (**B**) Dulaglutide titer accumulation. Dulaglutide titer was measured by Protein A analytical microcolumn.

**Figure 10 pharmaceuticals-18-01896-f010:**
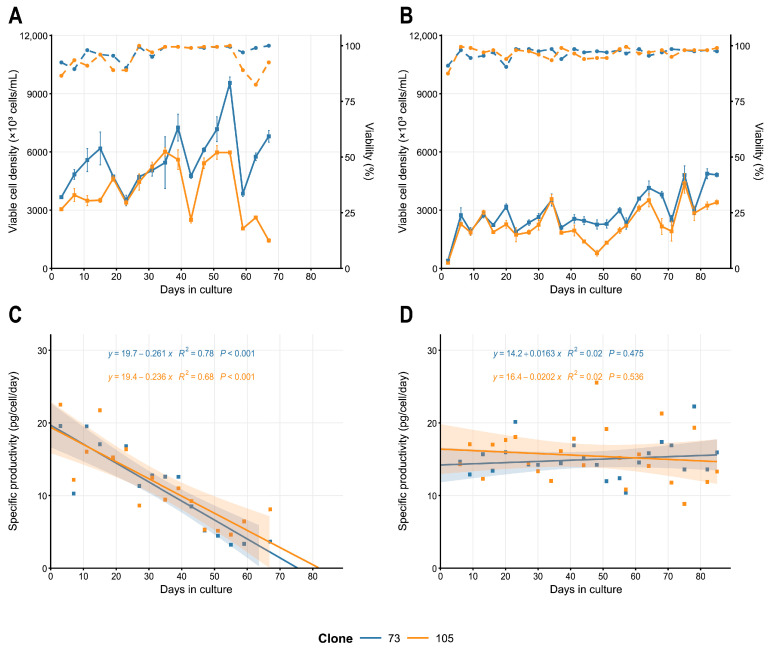
Cell dynamics (**A**,**B**) and specific productivity trends (**C**,**D**) for the main (#73) and reserve (#105) clonal cell lines producing dulaglutide during the long-term culturing in EmCD CHO 101 (**A**,**C**) and Pro CHO5 (**B**,**D**) media. VCD—solid lines; viability—dashed lines. Dulaglutide quantification was performed by ELISA, *n* = 2.

**Figure 11 pharmaceuticals-18-01896-f011:**
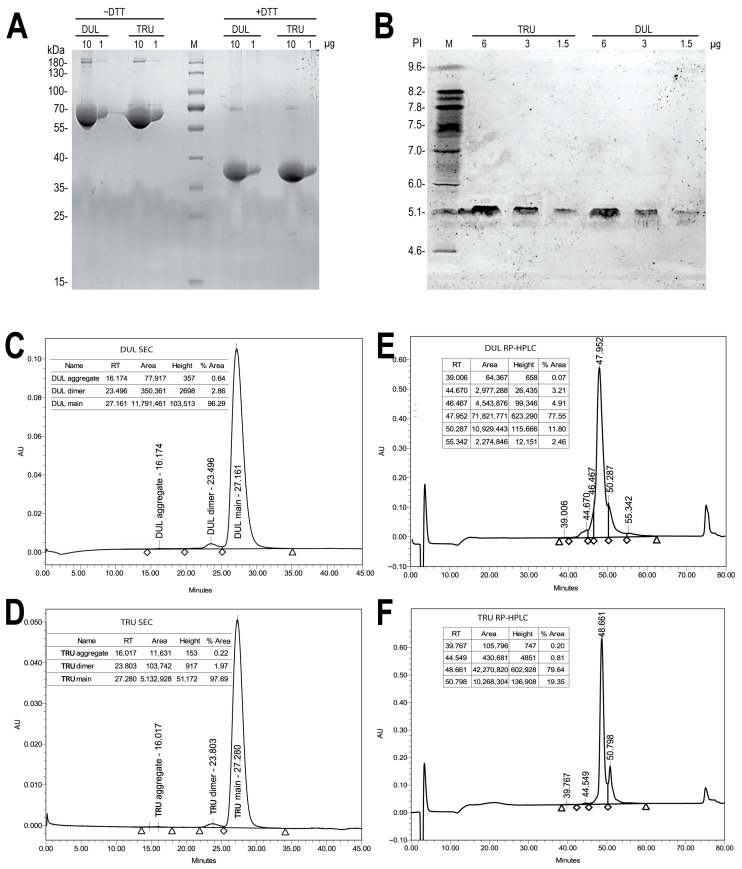
Characterization of affinity-purified dulaglutide. (**A**) SDS-PAGE in reducing and non-reducing conditions. (**B**) Isoelectric focusing. (**C**,**D**) SEC-analysis on a Superdex 200 column. (**E**,**F**) RP-HPLC-analysis. DUL—test sample; TRU—reference dulaglutide drug sample.

**Figure 12 pharmaceuticals-18-01896-f012:**
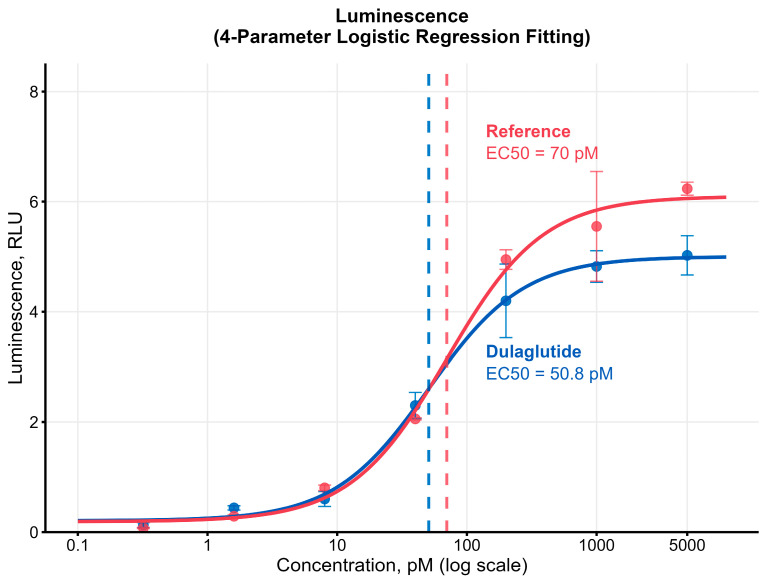
Dulaglutide bioactivity measured in vitro using GLP-1R-expressing HEK293 cell line in comparison with reference original product (Trulicity). Luminescence signal increase (RLU) under varying agonist concentrations. One representative experiment out of three is shown.

**Table 1 pharmaceuticals-18-01896-t001:** Relative dulaglutide glycoforms abundance as assessed using MS. DUL—test sample; TRU—reference drug Trulicity.

DUL			TRU		
Number	MW, Da	Relative Abundance	Number	MW, Da	Relative Abundance
1	31,286.4645	55%	1	31,286.3724	75%
2	31,448.0186	14%	2	31,448.4705	15%
3	31,366.412	13%	-	-	-
4	31,528.2998	7%	-	-	-
5	31,609.9932	3%	3	31,610.6656	3%
-			4	31,140.7391	3%
6	31,489.9405	3%	5	31,488.6724	2%
7	31,059.0143	2%	-	-	-
8	31,083.534	1%	-	-	-
9	31,759.2865	1%	-	-	-
10	31,569.8966	1%	-	-	-
-	-	-	6	31,377.2315	1%
-	-	-	7	31,545.7189	1%

## Data Availability

The original contributions presented in this study are included in the article/Appendix A. Further inquiries can be directed to the corresponding author.

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
