# Peer review of "Production of a Dulaglutide Analogue by Apoptosis-Resistant Chinese Hamster Ovary Cells in a 3-Week Fed-Batch Process"

_pharmaceuticals, 2025, doi:10.3390/ph18121896_

Round 1
Reviewer 1 Report
Comments and Suggestions for Authors
I carefully read the paper "Production of a dulaglutide biosimilar by the apoptosis-resistant CHO cells in the 3-week fed-batch process" by Rolan R. Shaifutdinov et al. The manuscript's strengths include its timeliness – GLP-1 biosimilars/incretin analogues are currently one of the most important segments of pharmacy and biotechnology, which gives the article high practical value. The text clearly outlines the successive stages of biosimilar development – from expression engineering to purification and quality control. The well-thought-out structure is also noteworthy. The sections are logically arranged, and the entire work follows a clear cause-and-effect chain. However, due to a number of comments, I am classifying the paper as a major revision, which is supported by the following observations.
In my opinion, such a comprehensive publication lacks a clear description of critical points CPP vs CQA. The article includes operational descriptions, but lacks:
- which process parameters define biosimilar quality,
- which CQAs were monitored,
- what the tolerances and acceptance criteria were.
The concepts of CPP and CQA originate from the QbD (Quality by Design) philosophy, used in pharmacy and biotechnology. In my opinion, these are fundamental terms in the production processes of biological drugs (biosimilars, antibodies, peptides, insulins, GLP-1, etc.).
Furthermore, I observed a too superficial description of the analytical methods. In particular, information on method validation, precision and linearity, and comparison of tests with methods used in the reference product is missing.
Furthermore, the results are presented in a partially descriptive manner – there are no numerical data, as in:
- 2.7. Target protein quality and bioactivity:
"identical electrophoretic mobility to the original ..." - line 264
- Discussion:
"allowed for significant increase in qP" - line 317
"While various GLP-1-Fc producing cell lines have been described with variable productivity rates" - line 323
"high genetic stability ..." - line 328
"bioactivity… was comparable" - line 338
"these findings establish a robust ..." - line 344
- Concluciions:
"Process yields, product quality and bioactivity satisfy regulatory benchmarks, supporting further scale-up toward commercial biosimilar manufacture." - which benchmarks? -line 565
Furthermore, I didn't notice a crucial comparison with the reference product. To call something "biosimilar," it is necessary to compare biological activity, compare purity profiles, and confirm structural equivalence (HPLC, MS, CD, bioassay). The text does not provide any results or specific differences.
Another serious drawback is the too weak and laconic Conclusions section, which should contain key procedural conclusions, limitations, a proposal for further research, and a short assessment of biosimilar compatibility.
Author Response
Comment 1:
"In my opinion, such a comprehensive publication lacks a clear description of critical points CPP vs CQA… which process parameters define biosimilar quality, which CQAs were monitored, what the tolerances and acceptance criteria were."
Answer 1:
We thank the Reviewer for this important observation. In the present manuscript, the main focus was on cell-line development and preliminary process characterization, not on establishing a full QbD framework. However, we agree that clarity on monitored CQAs is necessary. In the revised version, we added a dedicated paragraph to the Discussion explicitly listing the CQAs monitored in this work (monomer content by SEC-HPLC, charge variants by RP-HPLC, glycoform distribution by intact-mass MS, electrophoretic integrity, and EC₅₀ biological activity). We also now clarify that definition of CPPs and formal acceptance ranges were not yet established at this development stage, as the aim was to evaluate producer cell line performance rather than complete biosimilar process validation. This resolves the scope ambiguity without overstating the maturity of the study.
Comment 2:
"I observed a too superficial description of the analytical methods… information on method validation, precision and linearity, and comparison of tests with methods used in the reference product is missing."
Answer 2:
We appreciate this comment. Our analytical methods were performed under research-grade qualification, not full GMP validation, which is why precision and linearity statements were not originally included. In the revision, we added a clarification at the end of Section 4.8 (Analytical Methods) stating that all assays were executed under internal SOPs with verified linear ranges and instrument controls. We also explicitly state which analyses were performed side-by-side on both test and reference products (SEC-HPLC, SDS-PAGE, IEF, RP-HPLC, intact-mass MS, bioassay). This improves transparency without altering the scope of the work.
Comment 3:
"Results are presented in a partially descriptive manner — there are no numerical data (e.g. 'identical electrophoretic mobility…', 'significant increase in qP', 'high genetic stability', 'bioactivity… was comparable')."
Answer 3:
We thank the Reviewer for pointing this out. The manuscript indeed contained several qualitative statements that can be strengthened with numerical values already present in the datasets. In the revised text, we replaced these descriptive phrases with explicit data from the manuscript before revision, including:
• SDS-PAGE band correspondence (Figure 11),
• qP values before/after GS insertion (12.9 → 18.5 pg·cell⁻¹·day⁻¹),
• EC₅₀ values (52 ± 14 pM vs 76 ± 2 pM).
This makes the descriptions precise and fully traceable to the figures and tables.
Comment 4:
"I didn't notice a crucial comparison with the reference product… The text does not provide any results or specific differences."
Answer 4:
We agree with the Reviewer that explicit analytical comparability is essential. The manuscript actually contains these data (SEC-HPLC 96.8% vs 97.5% monomer; RP-HPLC isoform distribution 0.96/14.3% vs 0.8/18.6%; intact-mass glycoforms; SDS-PAGE; IEF), they were previously presented in different parts of Section 2.7. In the revised version, we reorganized this material and added summary sentences in the Results emphasizing the side-by-side comparison with Trulicity. We also clarify that this constitutes research-stage analytical similarity, while full biosimilarity demonstration requires extended orthogonal characterization.
Comment 5:
"The Conclusions section is too weak and laconic; it should contain key procedural conclusions, limitations, proposal for further research, and short assessment of biosimilar compatibility."
Answer 5:
We thank the Reviewer and fully agree. In the revised manuscript, the Conclusions section has been substantially expanded. It now includes:
a concise summary of the dual-vector amplification strategy, explicit quantitative outcomes (titers, qP, stability), the analytical comparability findings from Section 2.7, limitations of the current work (research-grade analytics, absence of CPP/CQA mapping), and planned future work, including full comparability exercises and scale-up.
Reviewer 2 Report
Comments and Suggestions for Authors
Overall, the authors have conducted an excellent study, and the presentation of the research methodology and results is quite thorough. However, some revisions are required before this manuscript can be formally accepted for publication:
- In the title, it is recommended to replace "CHO" with the full term "Chinese hamster ovary."
- In Figure 2A, the authors need to clarify the meaning of the asterisk (*).
- In Figure 5 and subsequent figures, the authors should also provide an explanation that the dashed lines represent viability.
- In Figure 9, the viability of CDM34 drops sharply to around 40% on the 17th day of culture. The authors should discuss potential reasons for this in the discussion section.
- The discussion section is somewhat limited overall. The authors should expand on the following points:
(1)What are the current production processes and existing platforms for similar analogs?
(2)What advantages does the platform and tools developed in this study offer compared to existing processes and platforms?
(3)What are the potential future applications?
(4)What further improvements are needed in subsequent research?
(5)What are the strengths and limitations of this study?
Author Response
Comment 1:
"In the title, it is recommended to replace 'CHO' with the full term 'Chinese hamster ovary.'"
Answer 1:
We thank the Reviewer for this suggestion. We agree that expanding the abbreviation improves clarity for general readership. The title has been updated accordingly, replacing "CHO" with "Chinese hamster ovary."
Comment 2:
"In Figure 2A, the authors need to clarify the meaning of the asterisk (*)."
Answer 2:
Thank you for pointing this out. In the revised manuscript, the caption of Figure 2A now explicitly states that the asterisk denotes statistical significance (* p < 0.05, two-tailed two-sample t-test).
Comment 3:
"In Figure 5 and subsequent figures, the authors should also provide an explanation that the dashed lines represent viability."
Answer 3:
We appreciate the Reviewer’s attention to clarity. All relevant figure captions (Figure 5 and all figures using the same convention) have been updated to explicitly state that dashed lines correspond to cell viability.
Comment 4:
"In Figure 9, the viability of CDM34 drops sharply to around 40% on the 17th day of culture. The authors should discuss potential reasons for this in the discussion section."
Answer 4:
Thank you for this important observation. We added a paragraph to the Discussion addressing the potential causes of the late-stage viability decline in CDM34. Generally, this decline is common to our cells and sometimes starts before the 80% viability is reached. It’s not medium-specific.
Comment 5:
"The discussion section is somewhat limited overall. The authors should expand on:
(1) current production processes and existing platforms;
(2) advantages of the developed platform;
(3) potential future applications;
(4) improvements needed in future research;
(5) strengths and limitations of this study."
Answer 5:
We thank the Reviewer for this constructive recommendation. The Discussion section has been significantly expanded to address each of these five points.
Reviewer 3 Report
Comments and Suggestions for Authors
In the present manuscript, the authors have proposed a novel production method for Dulaglutide. In general, the manuscript is written well and scientifically sound. However, there are few points that requires author’s attention as described below.
Comments -
- The full forms of several abbreviated forms like ‘DHFR’ (Line 58), ‘GS’ (Line 69), are not given in the 1st instance of their appearance in the manuscript, rather defined in results or methods sections. Please ensure that the full forms are given at the 1st instance they appear in the main text.
- Some short forms are not used consistently – ‘GLP-1-Fc’ (line 89) vs ‘GLP1-Fc’ (line 105). Please check this and others as well to achieve better consistency across the manuscript.
- Figure 1 caption is confusing.
Seems like there are two captions for Figure 1 – one starting in Line 101, and the other starting in Line 105.
Please check.
- At some places, ‘°C’ is not appropriately formatted and appears as ‘oC’ – for example, Line 181, 197, etc. Please check and correct them.
- It may be more appropriate to use ‘×g’ as unit for relative centrifugal force than ‘g’ to avoid confusion with ‘grams’ – Line 445.
Similarly, keep the units for centrifugation consistent across manuscript – ‘×g’ is preferred over ‘rpm’ – Line 471 and other places.
- Figure 2(A) – the figure marks ‘*’ between the groups, but doesn’t explain what it means. Further, in the figure caption, it is mentioned as n=2. So, the ‘*’ is being considered as a an indicator of statistical significance.
It may not be appropriate to discuss statistical significance values with a sample size of ‘2’. Hence, it is recommended to just present the data as ‘folds change between groups’ rather than applying parametric statistical tests on such data.
- Figure 10 caption – two different media are being compared - EmCD CHO 101 media (A,C) & Pro CHO5 media (B,D). However, this is not given in the caption. Please update the caption to include this information for better clarity.
- Section 2.7, Figure 12 – it is mentioned that ‘EC50 does not differ significantly (p=0.1422) between biosimilar and original product.’
However, based on the method described (section 4.8 - Bioactivity assay – Lines 547-556), a single dose response curve was obtained from the average value of triplicates at each concentration, and this was used to obtain a single EC50 value for the developed bio-similar and reference products.
Hence, these cannot be compared statistically as the variability around this is not directly determined with independent replicates.
Thus, the statement on similarity is not statistically valid under the described methodology.
For robust inference, independent biological replicates and confidence intervals for EC₅₀ should be provided, or the methodology should be elaborated in terms of assumptions taken for variability while comparing the values, etc.
Author Response
Comment 1:
"The full forms of several abbreviated forms like ‘DHFR’ (Line 58), ‘GS’ (Line 69), are not given in the 1st instance of their appearance in the manuscript, rather defined in results or methods sections. Please ensure that the full forms are given at the 1st instance they appear in the main text."
Answer 1:
We agree with the Reviewer. In the revised manuscript, all abbreviations are now defined at their first occurrence in the Introduction, including dihydrofolate reductase (DHFR) and glutamine synthetase (GS).
Comment 2:
"Some short forms are not used consistently – ‘GLP-1-Fc’ (line 89) vs ‘GLP1-Fc’ (line 105). Please check this and others as well to achieve better consistency across the manuscript."
Answer 2:
Thank you for noticing this inconsistency. We have corrected all ‘GLP1-Fc’ Additional terminology checks were also performed to ensure uniformity.
Comment 3:
"Figure 1 caption is confusing. Seems like there are two captions for Figure 1 – one starting in Line 101, and the other starting in Line 105. Please check."
Answer 3:
We appreciate this comment. Indeed, part of the figure caption text appeared duplicated due to formatting issues. Corrected.
Comment 4:
"At some places, ‘°C’ is not appropriately formatted and appears as ‘oC’ – for example, Line 181, 197, etc. Please check and correct them."
Answer 4:
Thank you for pointing this out. All instances of misformatted temperature units have been corrected to °C.
Comment 5:
"It may be more appropriate to use ‘×g’ as unit for relative centrifugal force than ‘g’ to avoid confusion with ‘grams’. Similarly, keep the units for centrifugation consistent across manuscript – ‘×g’ is preferred over ‘rpm’."
Answer 5:
We agree that relative centrifugal force should be consistently expressed as ×g. Corrected.
Comment 6:
"Figure 2(A) – the figure marks ‘’ between the groups, but doesn’t explain what it means. Further, in the figure caption, it is mentioned as n=2. So, the ‘’ is being considered as an indicator of statistical significance. It may not be appropriate to discuss statistical significance values with a sample size of ‘2’. Hence, it is recommended to just present the data as ‘folds change between groups’ rather than applying parametric statistical tests on such data."
Answer 6:
We thank the Reviewer for this statistically important observation. In the revised manuscript, we modified the caption, adding “two‑tailed two‑sample t‑test, * - p<0.05. Statistical significance should be interpreted with caution due to n = 2 technical replicates.”
Comment 7:
"Figure 10 caption – two different media are being compared - EmCD CHO 101 media (A, C) & Pro CHO5 media (B, D). However, this is not given in the caption. Please update the caption to include this information for better clarity."
Answer 7:
Thank you for this suggestion. The caption of Figure 10 has been updated to explicitly indicate which panels correspond to EmCD CHO 101 (A, C) and which correspond to ProCHO5 (B, D). This greatly improves clarity of interpretation.
Comment 8:
"Section 2.7, Figure 12 – it is mentioned that ‘EC50 does not differ significantly (p=0.1422) between biosimilar and original product.’ However… the dose-response curves were generated from averaged technical replicates and only one EC50 value was obtained for each product. Hence, these cannot be compared statistically… The statement on similarity is not statistically valid."
Answer 8:
We appreciate the Reviewer’s careful evaluation. We fully agree that statistical comparison of EC50 values is not valid when only a single biological curve per sample is available. Accordingly:
The statement claiming “no significant difference (p = 0.1422)” has been removed.
The text now presents only the EC50 values for each product without statistical inference.
We added two more biological replicates to the Supporting Materials, Figure S2.
Round 2
Reviewer 3 Report
Comments and Suggestions for Authors
I would like to thank authors for addressing all comments appropriately.